Association of triglyceride-glucose index with the risk of prostate cancer: a retrospective study

Li Tianqi 1
Zhou Yijie 1
Wang Jinru 1
Xiao Songtao 1
Duan Yajun 1
Li Caihong 1
Gao Yi 2
An Hengqing 9269735@qq.com 3
Tao Ning 38518412@qq.com 4
1 School of Public Health, Xinjiang Medical University , Urumqi , China
2 School of Traditional Chinese Medicine, Xinjiang Medical University , Urumqi , China
3 Department of Urology, The First Affiliated Hospital of Xinjiang Medical University , Urumqi , China
4 Department of Epidemiological Statistics, School of Public Health, Xinjiang Medical University , Urumqi , China
Evans D. Gareth
Electronic publication date: 2023 Nov 7
Publication date: 2023
Volume: 11
Electronic Location ID: e16313
Received 2023 May 31; Accepted 2023 Sep 27
Copyright: ©2023 Li et al.
Copyright year: 2023
Copyright holder: Li et al.
License: This is an open access article distributed under the terms of the Creative Commons Attribution License, which permits unrestricted use, distribution, reproduction and adaptation in any medium and for any purpose provided that it is properly attributed. For attribution, the original author(s), title, publication source (PeerJ) and either DOI or URL of the article must be cited.
License URL: https://creativecommons.org/licenses/by/4.0/

Keywords: Prostate cancer, Triglyceride-glucose (TyG) index, Hyperinsulinemia, Insulin resistance, Low-density lipoprotein, Total cholesterol

Funding: Key Project of Natural Science Foundation of Xinjiang Uygur Autonomous Region 2022D01D39 Xinjiang Uygur Autonomous Region Tianshan Talent Youth Top-notch Project 2022TSYCCX0026 Xinjiang Medical University’s 17th College Students’ Innovative Training Program Project S202210760063 This work was supported by the Key Project of Natural Science Foundation of Xinjiang Uygur Autonomous Region (Number: 2022D01D39); the Xinjiang Uygur Autonomous Region Tianshan Talent Youth Top-notch Project (Number: 2022TSYCCX0026); and the Xinjiang Medical University’s 17th College Students’ Innovative Training Program Project (Number: S202210760063). The funders had no role in study design, data collection and analysis, decision to publish, or preparation of the manuscript.

==============================
Background

Prostate cancer is the most common malignancy in men, and its incidence is increasing year by year. Some studies have shown that risk factors for prostate cancer are related to insulin resistance. The triglyceride-glucose (TyG) index is a marker of insulin resistance. We investigated the validity of TyG index for predicting prostate cancer and the dose-response relationship in prostate cancer in relation to it.

Objective

To investigate the risk factors of TyG index and prostate cancer prevalence.

Methods

This study was screened from the First Affiliated Hospital of Xinjiang Medical University and included 767 people, including 136 prostate cancer patients in the case group and 631 healthy people in the control group. The relationship between TyG index and the risk of prostate cancer was analyzed by one-way logistic regression, adjusted for relevant factors, and multi-factor logistic regression analysis was performed to further investigate the risk factors affecting the prevalence of prostate cancer. ROC curves and Restricted Cubic Spline were established to determine the predictive value and dose-response relationship of TyG index in prostate cancer.

Results

Blood potassium (OR = 0.056, 95% CI [0.021–0.148]), total cholesterol (OR = 1.07, 95% CI [0.792–1.444]) and education level (OR = 0.842, 95% CI [0.418–1.697]) were protective factors for prostate cancer, alkaline phosphatase, age, LDL, increased the risk of prostate cancer (OR = 1.016, 95% CI [1.006–1.026]) (OR = 139.253, 95% CI [18.523–1,046.893] (OR = 0.318, 95% CI [0.169–0.596]); TyG index also was a risk factor for prostate cancer, the risk increased with TyG levels,and persons in the TyGQ3 group (8.373–8.854 mg/dL) was 6.918 times (95% CI [2.275–21.043]) higher than in the Q1 group,in the TyGQ4 group (≥8.854) was 28.867 times of those in the Q1 group (95% CI [9.499–87.727]).

Conclusion

TyG index may be a more accurate and efficient predictor of prostate cancer.

Introduction

Prostate cancer is a common malignant tumor in men (Siegel, Miller & Jemal, 2018). According to the Global Cancer Report 2020, there will be an estimated 19.3 million new cases worldwide in 2020, of which 1,414,259 will be prostate cancer cases (i.e., prostate cancer accounts for 7.3% of all new cancer cases) (Deng, 2020). The incidence of prostate cancer in China is also showing a rapid and continuous increase, and the incidence of prostate cancer in Chinese men over 70 years of age ranks first in male genitourinary tumors (Han et al., 2013). Therefore, it is urgent to examine the relationship between prostate cancer and other modifiable risk factors. Several studies have shown that risk factors for prostate cancer include age, race, family history, insulin-like growth factors, lifestyle, diet, environmental and occupational exposures (Howlader et al., 2013; Jayadeyappa et al., 2011; Supit et al., 2013; Kwabi-Addo et al., 2010; Gennigens, Menetrier-Caux & Droz, 2006). Hsing et al. (2003) findings suggest that insulin resistance (IR) is associated with a higher risk of prostate cancer in Chinese men. Homeostasis model of assessment for insulin resistence index (HOMA-IR) is a widely used alternative indicator of IR in clinical practice (Pahkala et al., 2020). However, the measurement of insulin levels is mainly used as a screening indicator for diabetic patients and is not applicable to the general population (Yan et al., 2021). The triglyceride-glucose (TyG) index is a comprehensive index used to investigate the risk factors of insulin resistance (IR) and prostate cancer. It only requires the measurement of fasting blood glucose and fasting triglycerides, which is much easier and faster than measuring insulin. In fact, it has been shown that the TyG index is more accurate in predicting IR compared to IR surrogate markers (Vasques et al., 2011; Lee et al., 2014). In a study conducted in Xinjiang region, prostate cancer patients were studied using TyG index as a comprehensive index to investigate the risk factors of TyG index and prostate cancer. The study aimed to determine the predictive value of TyG index in prostate cancer and investigate the dose–response relationship between TyG index and prostate cancer using Restricted Cubic Spline model analysis. The effect of triglycerides on blood glucose and insulin was also analyzed, to investigate the interaction between triglyceride (TG), totalcholesterol (TC), and low density lipoprotein (LDL) and their joint contribution to prostate cancer, and to provide a reference for the prediction of prostate cancer.

Materials & Methods

Source

In this study, 300 patients with prostate cancer diagnosed pathologically by prostate puncture biopsy in the First Affiliated Hospital of Xinjiang Medical University between 2020 and 2022 were retrospectively collected, and 3,000 healthy people without chronic or underlying diseases were randomly selected as the control group during the same period. According to the inclusion and exclusion criteria, 136 patients with prostate cancer and 631 healthy people were finally selected, for a total of 767 patients.

Case group inclusion criteria: Patients who underwent prostate puncture biopsy and were pathologically diagnosed with prostate cancer; were able to read, understand and complete the informed consent form; and had complete general demographic characteristics and medical history.

Case group exclusion criteria: Exclude those with incomplete case information. Exclude those with hyperglycemia.

Inclusion criteria for the control group: Completely healthy people without cancer, underlying diseases, or chronic diseases.

Exclusion criteria for the control group: Exclusion of hypertension, diabetes mellitus, and chronic diseases of the prostate.

We calculated the sample size required for this case-control study by using PASS 11.0 software.

Sample size N = 644, sample size N1 = 44 for case group and N2 = 600 for control group. This study complies with the sample size calculation (Qiang & Maoyin, 2007; Xing, Yang & Yang, 2004).

Method

This study has been approved by the Ethics Committee of the First Affiliated Hospital of Xinjiang Medical University (20220308-166).

Information collection

The team collected information on general demographic characteristics and physical examination. General demographic characteristics include: education level, marital status, age, group, smoking, alcohol consumption, past medical history, etc. According to the relevant WHO definition, smoking: those who smoked ≥1 cigarette/day on average and for more than six months. Alcohol consumption: more than six months in a row with an average of ≥1 time/week and ≥50 g/time (He & Gao, 2009; World Health Organization, 1997).

The physical examinations were done by professionals from the Physical Examination Center of the First Affiliated Hospital of Xinjiang Medical University. All participants were dressed lightly and height and weight were measured with standardized instruments. At the same time, smoking, alcohol consumption, chronic diseases, and underlying diseases were assessed by paper-based questionnaires. All participants had blood collected early in the morning on an empty stomach for various physiological and biochemical examinations. Fasting blood samples were collected and analyzed for biochemical measurements. These included ALP, LDL, blood calcium, blood potassium, TC, and TG, which were measured using an analyzer.

Diagnostic criteria and definition of indicators

The triglyceride-glucose (TyG) index = (ln(fasting triglyceride (TG) (mg/dL) × fasting glucose (mg/dL)/2) index (Liu et al., 2021). According to the 2023 Journal of the Chinese Society of Hypertension blood pressure in healthy people is systolic blood pressure <140 mmHg and diastolic blood pressure < 90 mmHg (Akinyelure et al., 2022). The Chinese Journal of Circulation states that the normal value of LDL is <3.4 mmol/L (Zhu et al, 2016). The 2022 Chinese community physicians state that the normal value of TC is <5.2 mmol/L (Duan & Yao, 2022).

Statistical analysis

The data were analyzed using SPSS version 27, and measures conforming to normal variance chi-squared are expressed as x ¯±S, and those that did not conform to the normal distribution were expressed as median (interquartile spacing) (M(P25,P75)), the t-test or Mann-Whitney U rank sum test was used for comparison between groups. Qualitative data were expressed as the number of cases and composition ratio, and the chi-square test was used to compare whether there was a difference between patients with prostate cancer and those without prostate cancer. Factors influencing the association with prostate cancer were screened by logistic regression analysis. Logistic regressions were performed to evaluate the relationship between sleep duration and sleep quality with hypertension.

Restricted cubic spline were plotted to analyze the dose–response relationship between TyG index and prostate cancer. The test level α = 0.05.

Results

Baseline characteristics of the study subjects

A total of 767 study subjects were included, with 136 prostate cancer patients in the case group and 631 cases in the control group. Contrasting the marital status, ethnicity, smoking history, and BMI of the two groups, the differences were not statistically significant (P > 0.05); the education level, age, drinking history, and different TyG index levels of the two groups were compared, the differences were statistically significant (P < 0.001). Meanwhile, the alkaline phosphatase, low-density lipoprotein, blood calcium, blood potassium, and total cholesterol were statistically significant (P <  0.05). See Table 1.

Table 1 General information.

Variables	Prostate cancer patients	Non-prostate cancer patients	χ 2 / Z	P	
Education level/case (%)			21.822	<0.001	
Polytechnic school (including polytechnic school) and below	97 (73.10)	311 (49.30)			
Junior college and above	39 (28.70)	320 (50.70)			
Marital status/case (%)			0.329	0.566	
Married	132 (97.10)	620 (98.30)			
Other	4 (2.90)	11 (1.70)			
Ethnicity/case (%)			0.252	0.616	
The Han nationality	98 (72.10)	441 (69.90)			
Minority nationality	38 (27.90)	190 (30.10)			
Smoking history/case (%)			0.775	0.379	
Yes	36 (26.50)	191 (30.30)			
No	100 (73.50)	440 (69.70)			
History of drinking/case (%))			69.548	<0.001	
Yes	15 (11.00)	316 (50.10)			
No	121 (89.00)	315 (49.90)			
TyG Index Groups (mg/dl)			92.331	<0.001	
Q1 (<8.001)	9 (6.60)	182 (28.90)			
Q2 (8.001–8.373)	18 (13.20)	176 (27.90)			
Q3 (8.373–8.854)	34 (25.00)	156 (24.80)			
Q4 (≥8.854)	75 (55.10)	116 (18.40)			
Age/[M(P25, P75), years]	71.00 (64.25,78.75)	46.00 (43.00,48.00)	−17.956	<0.001	
BMI/[M(P25, P75), kg/m2]	24.00 (21.00,26.00)	23.88 (21.99,26.17)	−1.583	0.113	
TyG Index	8.92 (8.50,9.41)	8.29 (7.94,8.70)	−9.462	<0.001	
Alkaline phosphatase/[M(P25,P75), U/L]	71.12 (59.48,98.29)	62.80 (51.00,77.20)	−4.987	<0.001	
Low-density lipoprotein/[M(P25, P75), mmol/L]	2.30 (1.79, 2.81)	2.61 (2.30,2.90)	−4.595	<0.001	
Blood Calcium/[M(P25, P75), mmol/L]	2.27 (2.19,2.33)	2.30 (2.23,2.37)	−3.446	0.001	
Blood potassium/[M(P25, P75), mmol/L]	3.70 (3.40,3.99)	4.09 (3.88,4.30)	−10.091	<0.001	
Total cholesterol/[M(P25, P75), mmol/L]	4.01 (3.48,4.68)	4.56 (4.16,4.86)	−5.890	<0.001	

Influencing factors of prostate cancer prevalence

Univariate logistic regression analysis

The relationship between TyG index and the risk of prostate cancer prevalence was analyzed by univariate logistic regression, the results showed that the risk of prostate cancer in the TyGQ3 group (8.373–8.854 mg/dL) was 4.379 times higher than in the Q1 group (OR = 4.379, 95% CI [2.038–9.412]), and in the TyGQ4 group (≥8.854) was 13.075 times higher than in the TyGQ1 group (OR = 13.075, 95% CI [6.304–27.119]). See Table 2.

Table 2 Univariate and multivariate logistic regression analysis.

Variables	Univariate logistic regression analysis	Multivariate logistic regression analysis	
	P	OR	P	OR	
TyG index grouping (mg/dl)			<0.001		
Q2(8.001–8.373)	0.085	2.068(0.905–4.727)	0.049	3.124(1.006–9.697)	
Q3(8.373–8.854)	<0.001	4.379(2.038–9.412)	0.001	6.918(2.275–21.043)	
Q4 (≥8.854)	<0.001	13.075(6.304–27.119)	<0.001	28.867(9.499–87.727)	
Blood Calcium			0.056	0.106(0.011–1.060)	
Blood Potassium			<0.001	0.056(0.021–0.148)	
Total Cholesterol			0.660	1.070(0.792–1.444)	
Alkaline Phosphatase			0.003	1.016(1.006–1.026)	
Education Level			0.631	0.842(0.418–1.697)	
Drinking Grouping			<0.001	0.067(0.029–0.155)	
Age(≥47 years old)			<0.001	139.255(18.523–1046.893)	
LDL			<0.001	0.318(0.169–0.596)	

Multivariate logistic regression analysis

In order to further investigate the risk factors which influence the prevalence of prostate cancer,we used multivariate logistic regression analysis. The results were presented in Table 2: TyG index,age (LDL, blood potassium, total cholesterol, alkaline phosphatase, and education level were associated with prostate cancer. Among these indicators, blood potassium (OR = 0.056, 95% CI [0.021–0.148]), total cholesterol (OR =1.07, 95% CI [0.792–1.444]) and education level (OR = 0.842, 95% CI [0.418–1.697]) were protective factors for prostate cancer.However,alkaline phosphatase, age (LDL (increased the risk of prostate cancer (OR = 1.016, 95% CI [1.006–1.026],) (OR =139.253 (95% CI [18.523–1046.893]) (OR = 0.318, 95% CI [0.169–0.596]); TyG index also was a risk factor for prostate cancer, the risk increased with TyG levels,and persons in the TyGQ3 group (8.373−8.854 mg/dL) was 6.918 times (95% CI [2.275–21.043]) higher than in the Q1 group,in the TyGQ4 group (≥8.854) was 28.867 times of those in the Q1 group (95% CI [9.499–87.727]).

Predictive value of TyG index in prostate cancer

Prostate puncture biopsy pathology as the gold standard for the diagnosis of prostate cancer, the receiver operating characteristic (ROC) was employed to assess predictive value of TyG index,with the area under the curve (AUC) was 0.758 (95% CI [0.713–0.804]), the best cut-off point of TyG index was 8.497, and the sensitivity and specificity were 75.7% and 64.2%, respectively. The results showed that TyG index can efficiently predict the risk of prostate cancer. See Fig. 1.

Dose–response relationship between TyG index and prostate cancer

Adjust the relevant confounding factors, according to the maximum of R2 and Dxy value fit the optimal model, the optimal number of nodes for TyG index finally determined as 4. The restricted cubic spline (RCS) model was constructed to analyze the relationship between TyG index and prostate cancer prevalence. With the TyG index as the horizontal coordinate and the corresponding predictive value (OR) as the vertical coordinate, the upper and lower ranges represent the 95% confidence interval(CI). As shown in Fig. 1: there was a linear dose–response relationship between TyG index and the risk of prostate cancer prevalence (P < 0.05 for the general trend and P = 0.883 for the non-linearity relationship). The result of the graph illustrated a positive correlation between TyG index and prostate cancer prevalence, which means that the risk of prostate cancer increased with TyG index. See Fig. 2.

Figure 1 Predictive value of TyG index in prostate cancer.

Figure 2 Dose response relationship between TyG index and prostate cancer.

Discussion

Prostate cancer is one of the most common cancers in men and fourth in cancer-related mortality (Cozzi et al., 2022). A population-based study by Hsing et al., (2003) found that IR was associated with a higher risk of prostate cancer in Chinese men. The TyG index has been shown to be more accurate for IR prediction compared to IR surrogate markers (Vasques et al., 2011; Lee et al., 2014). The study found that higher levels of TyG index are a risk factor for prostate cancer development. The risk of prostate cancer in the TyG index Q4 group was 13.548 times higher than that in the Q1 group, suggesting that higher levels of TyG index are a risk factor for prostate cancer development. Therefore, this study applied ROC curve and restricted cubic spline model based on logistic regression model to analyze the prediction and dose–response relationship of TyG index on prostate cancer. The results showed that there was a linear dose–response relationship between TyG index and prostate cancer. The ROC curve in this study showed an area under the curve AUC of 0.758 (95% CI [0.713–0.804]), therefore, TyG index may be a more efficient and accurate marker of prostate cancer risk, and TyG index may be associated with prostate cancer prevalence.

TG and glucose are important indicators of TyG index. Elevated triglyceride levels increase the occurrence of oxidative stress and the production of reactive oxygen clusters, thus promoting tumorigenesis (Chen, An & Li, 2021). It can act directly on the pancreas and contribute to an increase in the concentration of oxygen free radicals in the body, leading to cellular DNA damage and also inducing tumors (O’Neill & O’Driscoll, 2015). It can also lead to an increase in blood glucose in the body, stimulating the pancreas and promoting an increase in insulin secretion (Jin & Quan, 2010). This will potentially lead to hyperinsulinemia. Studies by Pandeya et al. (2014), Prabhat et al. (2010), and Albanes et al. (2009) all showed a positive association between hyperinsulinemia and prostate cancer risk. This is due to the fact that when insulin enters the systemic circulation, it binds to insulin receptors on the cellular cell membrane, promoting cellular uptake of glucose, proteins, and small molecules, which facilitates cellular metabolism (Argiles & Lopez-Soriano, 2001). Insulin promotes mitogenic and growth-stimulating effects in the prostate, all of which may contribute to the development of cancer (Argiles & Lopez-Soriano, 2001). It has also been shown that hyperinsulinemia leads to the development of insulin resistance (IR) (Xing & Chen, 2022). At the same time, the development of insulin resistance worsens hyperinsulinemia (Xing & Chen, 2022), and therefore it is believed that insulin resistance and hyperinsulinemia together promote the development of prostate cancer. This could also corroborate our view that the IR surrogate marker TyG index is an important risk factor for prostate cancer.

Meanwhile, triglycerides are broken down in the body into fatty acids and glycerol, which undergo a series of biochemical reactions to produce acetyl coenzyme A, which serves as a raw material for cholesterol production (Zhou & Yao, 2018). The increase of triglycerides and cholesterol has a strong effect and association on the increase of blood glucose (Daboul, 2011). TG are closely related to blood glucose, and elevated TG are associated with significantly higher blood glucose. Elevated blood glucose also significantly affects TG. In a study by Uffe Ravnskov in 2022, it was shown that cancer cell growth may require the consumption of large amounts of TC (Ravnskov & McCully, 2022). A study by Wang et al. (2022) showed that TC promotes the proliferation of prostate cancer stem cells and causes prostate cancer progression. Thus, triglycerides and cholesterol work together synergistically to promote cancer cell proliferation and cancer cell progression.

This study also found that prostate cancer was associated with factors such as LDL and TC. The possible reason for this is that LDL transports TC from the liver to the blood and tissues throughout the body, which is why prostate cancer patients have lower levels of TC compared to the normal population. LDL leads to increased phosphorylation of STAT3 protein in prostate cancer cells and upregulates the levels of oncogenes controlled by this transcription factor (Jung et al., 2021). In addition, LDL enhances the proliferative and invasive potential of tumor cells (Jung et al., 2021).

Interestingly, back in 2012, in a study by Ying et al. (2012), it was concluded that LDL may have a positive correlation with IR, and in a study by Jung et al. (2021), it was also concluded that LDL was positively associated with prostate cancer. In the current study, a positive correlation between LDL and prostate cancer was also found, which also corroborates with their findings.

There are still some shortcomings in this study, firstly, this study is a single-center case-control study, so the sample size is limited to a certain extent, and in the future we will use a multi-center study in the Xinjiang region on the basis of this study, to further expand the sample size, and to collect more relevant factors for a more comprehensive study. Secondly, this study is a cross-sectional study, which cannot determine the causal relationship, and we will follow up through a cohort study to further explore the relationship between the dynamic changes of TyG index and prostate cancer during the follow-up process. Finally, because this is a case-control study, there is a certain bias, in the future there should be more strict inclusion and exclusion criteria when selecting research subjects, we will also be in the later stage through some statistical methods, such as stratification, matching, propensity score, analysis of covariance, etc. to further control the influence of confounding factors on the results of the interference, so that we can make the design more rigorous and the results more accurate!

In summary, LDL, total cholesterol, triglycerides and blood glucose are risk factors for prostate cancer, and triglyceride-glucose (TyG) index has a strong correlation with prostate cancer and may be a more accurate and efficient predictor of prostate cancer. Moreover, TyG index is easier to predict the risk of prostate cancer compared with IR markers. However, the exact mechanism needs to be further investigated in depth.

Conclusions

The TyG index may be a more accurate and efficient predictor of prostate cancer.

Supplemental Information

Supplemental Information 1 Flowsheet

Click here for additional data file.

Supplemental Information 2 Sample size

Click here for additional data file.

Supplemental Information 3 Raw data

Click here for additional data file.

Additional Information and Declarations

Competing Interests

Author Contributions

Human Ethics

Data Availability

The authors declare there are no competing interests.

Tianqi Li conceived and designed the experiments, performed the experiments, prepared figures and/or tables, authored or reviewed drafts of the article, and approved the final draft.

Yijie Zhou conceived and designed the experiments, performed the experiments, prepared figures and/or tables, authored or reviewed drafts of the article, and approved the final draft.

Jinru Wang analyzed the data, prepared figures and/or tables, and approved the final draft.

Songtao Xiao performed the experiments, prepared figures and/or tables, and approved the final draft.

Yajun Duan performed the experiments, prepared figures and/or tables, and approved the final draft.

Caihong Li performed the experiments, prepared figures and/or tables, and approved the final draft.

Yi Gao performed the experiments, prepared figures and/or tables, and approved the final draft.

Hengqing An performed the experiments, authored or reviewed drafts of the article, and approved the final draft.

Ning Tao performed the experiments, authored or reviewed drafts of the article, and approved the final draft.

The following information was supplied relating to ethical approvals (i.e., approving body and any reference numbers):

Ethics Committee of the First Affiliated Hospital of Xinjiang Medical University

The following information was supplied regarding data availability:

The raw measurements are available in the Supplemental Files.

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
