# Peer review of "Association of triglyceride-glucose index with the risk of prostate cancer: a retrospective study"

_PeerJ, doi:10.7717/peerj.16313_

## Round 0.1 · original submission · Major Revisions

Please address reviewer comments and in particular address the following in limitations: the power of the study and the adjustment for age and the appropriateness of the control group given their age distribution.

Reviewer 1 ·

Basic reporting

Li et al reported results from the study title "Association of triglyceride-glucose (TyG) index with the risk of prostate cancer". The findings suggested TyG index was associated with prostate cancer risk.

The authors applied logistic regression both uni and multivariate to obtain estimated risks. The authors also investigated other blood markers such as TC, TG etc.

The authors made a clear conclusion. I felt the paper could be benefitted from language check. There are some errors for example line 42, did the author means 2020 or 2022 (it was written as 20202). Line 53 need full stop. The authors used many abbreviations, these should be cited in full when first appeared.

Experimental design

The research question is original. The methodology section should describe power calculation as the study is quite small. Given that the authors extract information on pathological notes, would it be possible to classify prostate cancer cases into aggressive prostate cancer as this will be more informative in terms of comparison of this marker to distinguish between latent and aggressive form. Age is key factor in prostate cancer therefore age should be used for adjustment in the multivariate model. Mean age between cases and controls showed cases were far more older than controls.

Validity of the findings

As the study number is small. This should be discussed properly as it may affect the findings (large band 95% C.I). The authors should also discussed how this marker is stabilised over time particularly at one time point of measurement.

Reviewer 2 ·

Basic reporting

The paper was clear and well written. Literature cited was reasonable although a wider body of literature could have been referenced.

Experimental design

The design was standard but the implementation raised a series of key questions as to the sample size, the representativeness of the both the case series and the controls. Further parameters such as PSA and other factors would have been nice to know.

Validity of the findings

The 3 main issues are the power of the study and the adjustment for age and the appropriates of the control group given their age distribution. These factors fo raise concerns as to the validity of the findings presented.

Additional comments

An interesting marker but the problems of study implementation limit the confidence one can have in the validity of the results presented. It is also not clear whether the authors do have the information required to be able to improve the analyses further.

---

## Round 0.2 · Minor Revisions

Please respond to reviewer suggestions.

Reviewer 1 ·

Basic reporting

The authors have made significant changes to the article as requested. All raised points have been addressed adequately.

Experimental design

In the power calculation section, the authors could add more information as provided in their response. this would add a lot to the epidemiology paper as people would need to know what figures they applied to come up with to achieve such power. Or simple add this part in the supplement material.

Validity of the findings

This marker can be added to the literature.

Additional comments

Consistency of digits should be 2 or 3 throughout. There are some figures with 2 digits in the tables.

---

## Round 0.3 · accepted · Accept

Manuscript is ready for publication